# Towards the Next-Generation Disinfectant: Composition, Storability and Preservation Potential of Plasma Activated Water on Baby Spinach Leaves

**DOI:** 10.3390/foods8120692

**Published:** 2019-12-17

**Authors:** Mette Risa Vaka, Izumi Sone, Rebeca García Álvarez, James Leon Walsh, Leena Prabhu, Morten Sivertsvik, Estefanía Noriega Fernández

**Affiliations:** 1Department of Processing Technology, Nofima AS, NO-4021 Stavanger, Norway; mette.risa.vaka@nofima.no (M.R.V.); Izumi.Sone@Nofima.no (I.S.); rebeca_garciaalvarez@outlook.com (R.G.Á.); leena.prabhu@nofima.no (L.P.); Morten.Sivertsvik@Nofima.no (M.S.); 2Centre for Plasma Microbiology, Department of Electrical Engineering & Electronics, University of Liverpool, Liverpool L69 3BX, UK; J.L.Walsh@liverpool.ac.uk

**Keywords:** plasma activated water (PAW), plasma power, exposure time, PAW composition, PAW storage stability, fresh produce, baby spinach leaves, ready to eat, total bacterial counts, colour, shelf-life, disinfectant

## Abstract

Plasma activated water (PAW) has rapidly emerged as a promising alternative to traditional sanitizers applied in the fresh produce industry. In the present study, PAW chemistry and storage stability were assessed as a function of plasma operating conditions. Increasing plasma exposure time (5, 12.5, 20 min) and power (16, 26, 36 W) led to a significant drop in pH (2.4) and higher nitrates and nitrites levels (320 and 7.2 mg/L, respectively) in the PAW. Non-detectable hydrogen peroxide concentration, irrespective of the treatment conditions, was attributed to its instability in acidic environments and the remote PAW generation mode. pH, nitrates and nitrites levels in the PAW remained unaffected after two weeks at 4 °C. The potential of PAW for microbial inactivation and quality retention was demonstrated on baby spinach leaves. Rinsing steps influenced colour development during chilled storage to a greater extent than PAW treatment itself. About 1 log reduction in total bacterial counts (5 log CFU/g) was achieved through PAW rinsing, with no variability after eight days at 4 °C (typical shelf-life at retailers). Moreover, microbial levels on PAW-treated samples after storage were significantly lower than those on control samples, thus contributing to extended product shelf-life and reduced food waste generation.

## 1. Introduction 

Otherwise referred to as the fourth state of matter, manmade plasma is typically sustained via an electric discharge in a gas subset (e.g., room air); the partially or fully ionised air assembles a bunch of subatomic/molecular entities (reactive oxygen and nitrogen species, RONS) besides quanta of electromagnetic radiation (UV-photons, visible light), all co-existing as thermal or non-thermal plasmas upon thermodynamic equilibrium [1,2]. Although its commercial exploitation largely relies on upgraded surface features in advanced materials and a variety of usage domains in electronics, textiles, glass or paper, cold plasma (CP) induced at atmospheric-pressure and room-temperature has rapidly emerged as a value-added, niche opportunity for bio-based applications (e.g., food decontamination, functionalisation of food and food contact materials, agrochemical dissipation, mitigation of food allergens and anti-nutritional factors, edible oil hydrogenation, plant growth promotion, pest control, toxin removal, wastewater disinfection, cancer therapy, etc.), including mitigation of microbiological risks across food production systems [3]. Indeed, several studies have demonstrated the ability of CP to effectively inactivate a broad range of microorganisms, including spores [4], biofilms [5,6] and viruses [7,8], in an array of foods. Overall, CP stress mechanisms on bacterial cells have been associated to protein denaturation, lipid oxidation, mechanical stress on cell wall/membrane, cell shrinkage and cytoplasmic leakage, and DNA/RNA damage [9].

As CP treatment is limited to surface applications, which may negatively affect product colour, surface topography or bioactivity [10], a recent application of this technology to overcome such limitations relies on the activation of liquids (e.g., water) through their exposure to CP discharges (e.g., air plasmas), resulting in a cocktail of reactive oxygen and nitrogen species (RONS) often affecting the pH of the media (e.g., acidification). Plasma activated water (PAW) provides a series of advantages over direct CP treatment, e.g., dose control, ease of implementation, storage capacity, on/offsite generation, possibility for self-sanitation and reactivation, and sustainable production [11].

The type and concentration of reactive species in PAW depend on CP operating parameters (e.g., gas composition, plasma source, and power), exposure time, remote/direct generation mode, gap distance between the electrode and the liquid surface, etc. Hydrogen peroxide, hydroxyl ions, nitrates, nitrites and peroxynitrites have been reported as persistent species in PAW responsible for its antimicrobial activity [12,13,14], which is attributed to oxidative damage of the cell membrane, cell wall breakdown (intra-molecular bonds of peptidoglycan), cell shrinkage and cytoplasmic leakage, and DNA breakdown, besides mutagenic and cytotoxic damage [11].

Neither a chemical reagent nor a natural source, PAW is classified as purified water with promising applications in the food industry as e.g., curing agent, fertiliser, or postharvest sanitiser in the fresh produce industry [11,15]. Fresh-cut (minimally processed) vegetables are a rapidly developing class of convenience (ready-to-eat) foods and an important component of a healthy and balanced diet towards prevention of e.g., obesity, cardiovascular diseases, diabetes and cancer [16]. Minimal processing of fresh-cut vegetables is intended to improve product functionality without compromising microbial safety and quality/freshness attributes. However, major challenges facing the fresh produce industry are the rapid deterioration and limited shelf-life of such products as compared to whole items, due to their high respiration and transpiration rates, and their susceptibility to enzymatic and microbial degradation [17]. With worldwide per capita consumption of fresh produce being 20% to 50% short of the FAO/WHO recommended intake (400 g/d) [18], colour is particularly one of the most important quality attributes affecting consumer acceptability and eating experiences [19]. The discoloration of green leafy vegetables is the first visible symptom of senescence, which may compromise their economic value, and is attributed to enzymatic degradation of chlorophyll pigments during leaf processing and storage [20], while harvesting time may affect the initial colour of the leaves and its subsequent changes due to a variation in the level of the endogenous antioxidants [21]. Oxygen-reduced modified atmosphere packaging and rinsing with e.g., oxalic acid or high oxidation reduction potential (ORP) water have been shown to partly mitigate yellowing during storage. However, the high respiration rate of such products otherwise requires significant levels of oxygen in the package for quality preservation purposes, and chemical residues may also compromise toxicological safety [22]. Furthermore, green leafy vegetables such as spinach are typically linked to foodborne outbreaks, with irrigation/washing step and surface cross-contamination being acknowledged as major sources for microbial contamination [16,23], including bacteria (*Escherichia coli* O157:H7, *Salmonella* spp., *Listeria monocytogenes*, *Campylobacter* spp., *Shigella* spp.), viruses (norovirus, rotavirus, hepatitis A virus) and parasites (*Giardia*, *Echinococcus*, *Cryptosporidium*, *Entamoeba*, *Cyclosporidium*). Among them, the highest prevalence of foodborne diseases has been reported for norovirus/*Salmonella* and leafy greens consumed as raw salads [24,25]. PAW has recently gained growing interest as a sustainable, cost-effective alternative to current disinfectants based on artificial chemical cocktails [26]. Nevertheless, further studies on PAW composition, stability and preservation ability on representative products, are still required towards technology upscaling and industrial uptake.

In the present study, the chemical composition of PAW generated with a surface dielectric barrier discharge (SDBD) set-up was determined as a function of the CP power (voltage peak-to-peak) and exposure time, as well as its storage stability under relevant conditions for industrial settings (two weeks at 4 °C). Moreover, the potential of PAW for microbial disinfection, quality retention (colour) and shelf-life extension was assessed on baby spinach leaves after eight days of refrigerated storage.

## 2. Materials and Methods

### 2.1. Cold Plasma Generation System

The CP reactor, consisting of the powered and ground electrodes and a 1 mm thick quartz disc between them, was set up to generate a surface barrier discharge (SBD). Such a configuration was coupled to the lid of the treatment chamber (176 × 174 × 48 mm), with a total discharge area of 15 cm^2^. Since species with high oxidation potential are also short-lived, the gap distance between the SBD and the liquid (remote generation mode) limits the flux reaching the water surface and thus, efficiency drops. However, placing electrodes in contact with the liquid can also lead to further contamination and accelerated electrode wear. For 100 mL treatment volume, the gap distance between the liquid surface and the electrode was 44.8 mm (3.2 mm water column). The CP generating source produced a sinusoidal signal at a frequency of 12 kHz. Plasma power of 16, 26 or 36 W, corresponding to voltage peak-to-peak values of 9, 10 and 11 kV, respectively, was determined using the mean of the product between the applied voltage and current over 200 cycles. The system operated at atmospheric pressure, with room air as the plasma-inducing gas.

### 2.2. Determination of PAW Reactive Species

Spectroquant^®^ test kits (Merck, Oslo, Norway) were used for spectrophotometric determination (Shimadzu UVmini-1240-UV-VIS, Shimadzu, Tokyo, Japan) of nitrates (#109713; analogous to DIN 38405-9) and nitrites (#114776: analogous to EPA 354.1, APHA 4500-NO2-B, and DIN EN 26 777) at 340 and 525 nm, respectively. Hydrogen peroxide was determined with the titanium sulphate colorimetric method at 407 nm [27]. As nitrites may interfere with the determination of hydrogen peroxide due to the acidic PAW environment, the protocol was slightly modified by adding sodium azide 60 mM to the acidic samples prior mixing with the titanium sulfate reagent, to scavange reactive species (nitrites are reduced into molecular nitrogen) that could otherwise react with hydrogen peroxide [12,28]. Calibration curves for nitrates (0.5–100.0 mg/L), nitrites (0.01–3.00 mg/L) and hydrogen peroxide (0.1–5.0 mM) were determined in triplicate on independent days with distilled water as blank.

After exposure to CP and prior to the quantitative determination of reactive species, PAW samples were tempered at 15–20 °C, which is the optimal range for analytical determination, and appropriate dilutions were prepared with distilled water at room temperature.

### 2.3. PAW Chemistry and Storage Stability

CP activation experiments were conducted with distilled water under standard/constant conditions of magnetic stirring (500 rpm), initial volume (100 mL), pH (no adjustment, ≈5.3–6.0) and temperature (room temperature, ≈19–21 °C). A full factorial design was implemented (3 levels × 3 levels; at least in triplicate on independent days) to assess the effect of the plasma power (voltage peak-to-peak) [16 W (9 kV), 26 W (10 kV) and 36 W (11 kV)] and exposure time (5, 12.5 and 20 min) on the concentration of reactive species (see Section 2.2), pH and oxidation reduction potential (Mettler Toledo SevenGo Pro pH/ion meter, Mettler Toledo, Oslo, Norway) and immediate temperature reading (Raytek MiniTemp FS Infrared Food Thermometer, Raytek, Oslo, Norway) in PAW samples. Such a range of operating conditions was selected upon a preliminary semi-quantitative screening with MQuant^TM^ (Merck, Norway) test strips for nitrites (#110022) and nitrates (#110020). 

PAW stability (concentration of reactive species and pH) after 24 h, 1 week and 2 weeks of storage at 4 °C was determined (in triplicate) for those tests carried out at 20 min CP exposure time (for the three values of plasma power), since, irrespective of the plasma power, such an activation period resulted in the lowest pH and the highest level of reactive species and oxidation reduction potential in the PAW (see Section 3.1), thus representing the most promising conditions for further food decontamination trials and industrial implementation.

### 2.4. Effect of PAW on the Microbial Safety and Quality of Baby Spinach Leaves

Fresh-cut and unwashed baby spinach leaves (*Spinacea oleracea*) were kindly supplied in bulk by a local wholesaler (Oslo, Norway) under refrigerated transport conditions. On the day of reception, the stems were removed with a sterile scalpel on disposable Petri dishes, and 5 g samples were placed in sterile stomacher bags (Grade, UK) and stored overnight at 4 °C. Treatments were conducted in triplicate with either PAW (generated at 11 kV for 20 min on the 3 days prior to the treatment) or distilled water at 4 °C (to favour PAW stability), as follows: 5 g samples were placed into a high-density polyethylene (HDPE) container (RPC Bebo Pack, Norway) filled in with 100 mL of either PAW or distilled water, and rinsed for 2 min at 120 rpm on a laboratory rocker (Labnet ProBlot™ Rocker 25, Labnet, Oslo, Norway). The treatment was subsequently repeated to mimic industrial settings and the samples were then centrifuged in a home salad spinner for 5 min. Based on the available literature body, PAW generation conditions (activation time and plasma power) yielding the lowest pH and the highest levels of reactive species, oxidation reduction potential and stability during storage (see Section 3.1 and Section 3.2) were selected for product disinfection trials, towards further industrial implementation.

Surface colour of at least three baby leaves from each batch (CIE L*a*b*) was measured by image analysis (DigiEyeTM, VeriVide Ltd., Leicester, UK) in an illumination cabinet that ensures a uniform lighting (standard daylight, 6400 K). A Nikon camera D80 with a Nikkor lens was used. The colour of each sample was measured at a circled area at three different locations on the leave surface using the DigiPix colour measurement software (VeriVide, UK).

The remaining leaves from each batch were weighed and placed in a sterile stomacher bag with 40 mL peptone salt water (0.1% *w/v* peptone and 0.85% *w/v* NaCl) and homogenised (Smasher blender, Biomérieux, France) for 2 min. One mL aliquot of the cell suspension was sampled, serially diluted in peptone saltwater and spread onto Plate Count Agar (PCA) plates in triplicate by using an IUL Eddy Jet Spiral Plater (IUL Instruments, Barcelona, Spain). The plates were incubated at 37 °C for 48 h prior to enumeration of total viable counts. Microbiological and colour analysis were also conducted on untreated baby spinach leaves from three independent replicates.

After centrifugation, samples treated with either PAW or distilled water (three independent replicates) were packed in air (Webomatic SuperMAX, Bochum, Germany) and stored at 4 °C. The headspace was measured in three independent replicates with a Checkmate 9900 analyzer (PBI-Dansensor, Ringsted, Denmark) as follows: a 20 mL aliquot of the headspace gas was collected with a syringe after intrusion of the cover film, which was sealed with a foam rubber septum (Nordic Supply, Skodje, Norway) to avoid the diffusion of false atmosphere into the gas analyser. Microbiological and colour analysis were conducted after 8 days of chilled storage (typical product shelf-life at retailers), also on packed untreated baby spinach leaves kept at 4 °C for the same time period.

### 2.5. Statistical Analysis

Statistical analyses were performed using the SPSS statistics software package version 25 (SPSS Inc., Chicago, IL, USA). Two-way ANOVA was used to examine the significant effects of treatment and storage time and its interaction. One-way ANOVA and Tukey post hoc test were performed to test simple main effects and for the pair-wise comparison, respectively. The significance level cut-off was set at 95% (*p* ≤ 0.05).

## 3. Results and Discussion

### 3.1. PAW Composition as a Function of Plasma Power and Exposure Time

Figure 1, Figure 2, Figure 3 and Figure 4 display the results (average and standard deviation of three independent replicates) corresponding to the effect of plasma power (voltage peak-to-peak) and plasma exposure time on the pH, temperature (immediate reading), and nitrates and nitrites levels in PAW, respectively. Hydrogen peroxide has not been detected in the PAW samples (no significant differences with respect to the blank), irrespective of the treatment conditions, albeit it is acknowledged as one of the persistent species in the PAW commonly associated to its antimicrobial activity. The absence of this compound in PAW has been attributed in the literature to the rapid decomposition of hydrogen peroxide by nitrites under acidic conditions [12,28]. Thus, Shainsky et al. (2012) [29] have reported a decrease in hydrogen peroxide levels from 2000 to 10 mg/L just 10 min after CP treatment (Dielectric Barrier Discharge, DBD, system, 100 μL, 1.5 mm gap, 30 s) due to the low pH. However, in the present work, no significant differences were found in the absorbance at 407 nm when sodium azide was added to the samples, also exhibiting similar values to those obtained for the blank in distilled water (neutral pH). Moreover, the remote PAW generation mode (i.e., electrode not immersed in the water) and the relatively long exposure times in the present study may have also affected the availability of hydrogen peroxide. For instance, levels up to 50 mg/L have been reported in plasma jets (direct treatment) after 3 min exposure (20 mm gap), although the treatment volume was only 500 μL [30]. Indeed, the initial liquid volume and depth (100 mL, 3.2 mm), and the gap distance between the electrode and the liquid surface (44.8 mm) have also been acknowledged to play an important role in interface reactions and thus, in the chemical composition of PAW. For instance, hydrogen peroxide has been detected (3.4 mg/L) with a DBD system operating at 5 kV for 20 min with similar geometry (42 mm electrode-liquid gap, 2.5 mm liquid depth) but significantly lower treatment volume (10 mL) than that in the present study (dilution effect) [31]. Likewise, hydrogen peroxide levels of 2000 mg/L have been reported in a DBD system after 90 s exposure, but the volume (100 μL) and electrode-liquid gap (1.5 mm gap) were much smaller than the values used in this work [29]. Thus, the most effective PAW generation system with regards to hydrogen peroxide production has been reported to utilise water droplets that may sequester this compound in the liquid, thus suppressing decomposition reactions by radicals from the gas and at the interface [32].

From Figure 1, it can be noticed that the pH of PAW samples decreased as the exposure time and voltage peak-to-peak (plasma power) increased. At constant peak-to-peak voltage, the pH drop was statistically more pronounced at shorter exposure times (between 5 and 12.5 min), as no significant differences were found between 12.5 and 20 min exposure. At constant exposure time, the acidification of the media was statistically similar for peak-to-peak voltages of 9 and 10 kV, with more pronounced pH drop at 11 kV. In general, the RONS in the CP discharge have been reported to diffuse/dissolve into the water during the exposure and react with the water molecules giving rise to a cocktail of chemical species, whose generation is subjected to the release of hydrogen ions causing a drop in pH (up to 2.4 ± 0.1 in the present study), depending on the operating conditions [12,33]. For instance, nitrites and nitrates are formed in the PAW through the dissolution of nitrogen oxides (NO_x_) formed in the air plasma by gas-phase reactions of dissociated N_2_ and O_2_. The dissolution of NO_x_ in water also produces H^+^ ions (drop in pH). Likewise, hydrogen peroxide is formed in the PAW through the recombination reaction of OH• radicals produced by plasma at the gas/liquid interface. A comprehensive overview of species creation reactions is provided by Lukes et al. (2014) [12] and Thirumdas et al. (2018) [11]. Such a correlation between pH evolution and RNS levels has also been observed in the present work (Figure 3 and Figure 4). Furthermore, PAW ORP values, which provide a rapid and single-value assessment of the disinfection potential [13], increased with higher plasma power and activation time, with values ranging between 200 and 292 mV for extreme operating conditions (9 kV/5 min and 11 kV/20 min).

To the knowledge of the authors, pH 2 is the lowest value reported in the literature for PAW, where 10 mL pure deionized water was treated for 15 min at 17 kV with a DBD system [29]. The acidification levels after CP treatment also depend on the water source, i.e., whether distilled or tap water are utilised, and the buffering capacity of the tap water [34]. Distilled and (pure) deionised water are typically used for PAW generation due to its reproducibility in composition, unaffected by seasonality and occasional variations in the public water supply. For tap water at pH 7.5, only PAW generated at high voltage peak-to-peak (10 and 11 kV) and long exposure times (12.5 and 20 min) exhibited significant acidification (results not shown). However, targeting agronomy applications, Judee et al. (2018) [34] have used tap water at pH 7.8 for PAW generation (50 mL) in a DBD system operating at 12 kV, with a negligible variation in pH after 30 min treatment, despite the high concentration of RONS. Finally, the initial pH of the water has been adjusted to alkaline conditions in a few studies targeting improved curing and preservation properties [15,33,35]. For 11 kV and 20 min, no significant differences were found in nitrates and nitrites levels (results not shown), when the initial pH was adjusted to alkaline values with sodium pyrophosphate buffer 1% *w/v* (pH 9.7), despite the higher pH of PAW samples (≈9).

Figure 2 shows that the temperature of PAW samples (immediate reading after CP treatment) increased towards CP exposure time and voltage peak-to-peak, due to a thermal effect attributed to the SDBD electrode. A maximum temperature of 30.9 ± 1.2 °C (average initial temperature of 19 ± 0.7 °C) was reached for the most severe conditions of plasma power and exposure time, and no variations in the volume of water before and after activation were recorded. This value corresponds to relevant literature, where PAW temperatures up to 38.1 °C have been reported with an Ar/O_2_ plasma microjet on 20 mL distilled water (10 mm gap) operating at 12 W for 20 min [36]. Distilled water at room temperature has often been used for PAW generation, although this factor might affect solubility of RONS and thus, PAW composition. In this sense, nitrates levels remained unaffected when PAW was generated at 11 kV for 20 min with water initially tempered at 4 and 10 °C (results not shown), although slight differences were observed in the temperature after CP treatment (≈28 °C), nitrites levels (≈28 mg/L), and pH (≈3.6), the latter attributed to the higher concentration of nitrites.

Regarding the effect of CP power and exposure time on the RNS in PAW (Figure 3 and Figure 4), nitrates levels increased significantly towards both variables, with 11 kV voltage peak-to-peak (36 W plasma power) and 20 min of plasma exposure yielding the highest concentration (320 ± 47.8 mg/L). Nitrites levels followed a similar evolution with respect to the treatment duration and plasma power, although the maximum concentration achieved in PAW samples (7.2 ± 3.8 mg/L) was significantly lower than the corresponding one for nitrates. This behaviour has been attributed to the instability of nitrites under acidic conditions, which form nitrous acid, later decomposed into nitrates and nitrogen oxide [37,38]. Most studies in the literature have also reported higher concentrations of nitrates (e.g., up to 220 mg/L [34]), as compared to nitrites (e.g., only traces [36]). As nitrites remain stable under alkaline conditions, Jung et al. (2015) and Yong et al. (2017) [15,33] have favoured the formation of nitrites over nitrates (782 vs. 358 mg/L, respectively, after 120 min exposure in a SDBD system) by adjusting the initial pH of the water with sodium pyrophosphate buffer 1% *w/v* (pH 9.7), targeting curing applications. The energy density, i.e., the energy consumed for the creation of 1 mol of RNS, ranged between 7.5 and 8.1 × 10^4^ kJ/mol for extreme operating conditions (16 W/5 min and 36 W/20 min, respectively).

In the present study, nitrites levels for voltages peak-to-peak of 9 and 10 kV were practically negligible at any exposure time. Despite the relative variability among replicates, nitrites concentration at 11 kV significantly increased after 12.5 and 20 min treatment. However, often in literature [35,39,40] nitrites levels in the PAW have been reported to decrease with the exposure time to the plasma. For instance, Kojtari et al. (2013) [35] have reported non-detectable nitrites levels after 2 min treatment.

Likewise to hydrogen peroxide, the initial volume of water also plays a decisive role in RNS levels. Different volumes of water (ranging from 1 mL to 1.6 L) have been assayed in the literature, as well as gap distances between the plasma electrode and the liquid surface (ranging from 5 mm to 5 cm). For 11 kV and 20 min, an increase in the treatment volume from 100 to 500 mL (44.8 vs. 32 mm gap) significantly affected the pH (≈7.0), temperature (≈22 °C) and nitrates levels (≈40 mg/L) in the PAW as compared to 100 mL, although nitrites levels remained unaffected, most likely due to the neutral pH (results not shown). As magnetic stirring during the activation was kept constant (500 rpm) for both 100 and 500 mL initial volumes to avoid the contact of the water with the SDBD set up, the diffusion of RONS in the water may still be enhanced by adjusting this factor. However, for a similar SDBD system and 500 mL water, significantly higher nitrites and nitrates levels as compared to this study, have been reported at 300 rpm [15], although the treatment time (2 h) and initial pH (adjusted to 9.7) may have contributed as well. On the other hand, static conditions for PAW generation have been used in a few studies on microbial inactivation, although the water volumes and activation times were also much lower [29,30,31,35,39].

### 3.2. PAW Storage Stability

Figure 5, Figure 6 and Figure 7 display the results (average and standard deviation) corresponding to the stability of PAW (pH and nitrates and nitrites levels, respectively) generated at 9, 10 and 11 kV voltage peak-to-peak (16, 26, and 36 W plasma power) and 20 min of plasma exposure time, after 24 h, 1 week and 2 weeks of storage at 4 °C, with the aim to simulate conditions relevant for industrial settings. For all the conditions assayed, pH, nitrates and nitrites levels remained stable after 2 weeks of chilled storage (no statistically significant differences). However, a slight increase/decrease was observed in average nitrates/nitrites levels, respectively, which is attributed to the acidic pH of PAW, leading to the decomposition of nitrites into nitrates and nitrogen oxide [37,38].

A recent study assessing PAW bactericidal effect against *Staphylococcus aureus* as a function of PAW storage conditions (−80, −20, 4 and 25 °C, up to 30 days) has reported that the concentration of reactive species in PAW (hydrogen peroxide, nitrites and nitrates) decreased as the storage time increased and such a decrease was more pronounced at higher storage temperatures [41]. Traylor et al. (2011) [31] have also assessed the long-term antibacterial activity of PAW (7-day storage, temperature non disclosed). While hydrogen peroxide and nitrites levels decreased below the detection limit within two days, the pH remained constant over the whole period, and nitrates levels increased first rapidly and then more slowly over the following four days. On the contrary, Niquet et al. (2018) [42] have observed an increase in the hydrogen peroxide concentration and similar nitrates levels after 24 h storage at 4 °C, while nitrites were not detected before/after storage. According to Julak and Scholtz (2012) [43], the hydrogen peroxide content in samples exposed to a positive streamer discharge for 60 min and stored for 4 weeks under refrigeration decreased markedly, but still remained at 50 mg/L. However, only negligible amounts of hydrogen peroxide remained in the water exposed to a DC negative glow corona, which emphasizes the influence of the CP source in the chemical composition and storage stability of PAW. Within the plasma medicine field, Adachi et al. (2015) [44] have demonstrated the ability of plasma activated media (PAM) to suppress cell viability for at least 1 week when stored at −80 °C. In conclusion, the storage temperature and period, as well as the initial PAW chemistry and generation conditions, play an important role in the stability and biological activity of the PAW. Moreover, Kutasi et al. (2019) [45] have recently shown that PAW ageing critically depends on the initial concentration of the generated RONS, governed by the reactions detailed in Lukes et al. (2014) [12], which also defines the pH of the media. Such an effect of the initial concentration of reactive species has not been clearly observed in the present work for NO_3_, although it might explain the behaviour of NO_2_ during chilled storage.

### 3.3. Effect of PAW on the Microbial Safety and Quality of Baby Spinach Leaves

Figure 8 displays the results of the microbiological analysis (average and standard deviation of three independent replicates) of baby spinach leaves either untreated or processed with PAW and distilled water (control), after rinsing (0 d) and after 8 days of storage (air) at 4 °C (typical product shelf-life at retailers). Right after the rinsing, total bacterial counts on PAW-treated samples were significantly lower than those achieved on untreated samples, although statistically significant differences were not found with regards to control samples rinsed with distilled water. It is noteworthy that although the indigenous bacterial flora on the control samples was reduced after rising with distilled water, total bacterial counts after 8 days of incubation at 4 °C were significantly higher than the bacterial load remaining on both PAW-treated and untreated samples. Despite the statistical analysis not yielding significant differences, the average bacterial concentration in PAW-treated samples was lower than the levels in untreated samples. Thus, about 1 log reduction in total bacterial counts has been achieved through PAW rinsing, with microbial levels resulting unaffected (5 log CFU/g) after 8 days of storage at 4 °C.

There is a lack of consensus within the available literature body about the nature of PAW disinfection efficacy, which has been attributed to e.g., high oxidation reduction potential and electrical conductivity, low pH, nitrates, nitrites, peroxynitrites, peroxides (e.g., hydrogen peroxide), short-lived NO radicals and ROS (e.g., hydroxyl radical, singlet oxygen, superoxide anion), ozone and even metal nanoparticles released from the plasma electrode [13,28,29,36,41]. In the present work, the bactericidal activity is mainly attributed to RNS (NO_3_, NO_2_), low pH, and high oxidation reduction potential. However, the presence of other short-/long-lived species in the PAW might also have played an important role, as well as the plasma generation system. For instance, while high concentrations of hydrogen peroxide have recently been associated to argon discharges, high levels of nitrogen species and high variability of pH and electrical conductivity have been related to air discharges, as also observed in the present work [46].

The lightness (L* value), redness (a* value) and yellowness (b* value) of untreated, PAW-treated and control samples measured right after rinsing (0 day) and after 8 days of (air) storage at 4 °C are summarized in Figure 9 (average values and standard deviation of three independent replicates). Right after rinsing (day 0), no significant differences were found in the b* value among the three assayed conditions (untreated, PAW-treated and control). However, significant variability was observed for the a* (mean range: −12.6 and −14.3) and L* values (mean range: 36.2 and 38.7), likely indicating biological variability in the redness and lightness between the individual leaves analyzed. It is noteworthy that average a* and L* values for PAW-treated samples were closer to the corresponding ones for untreated samples, as compared to the controls. After 8 days of storage at 4 °C, no significant differences were found in the a* and b* values for PAW-treated and control samples. The L* value of the control samples was found to be similar to the untreated samples and significantly lower than that of the PAW-treated samples, but the average difference in the lightness of both treatments was limited to one unit, well within the range exhibited right after rinsing (day 0). On the other hand, the leaves subjected to either PAW or distilled water rinsing exhibited significantly lower and higher a* and b* values, respectively, than those for the untreated samples, due to relatively more pronounced yellowness and larger deviation from the initial greenness in the colour of rinsed samples as compared to untreated ones, as a result of storage. Such a difference is also attributed to the significant increase in greenness observed in the untreated samples at day 8, as compared to the colour obtained at day 0, characterized by significantly higher and lower a* and b* values, respectively. These results demonstrated that rinsing steps influenced colour development of spinach baby leaves to a greater extent than the PAW treatment itself during chilled storage. In fact, the yellowness of PAW-treated samples remained unaffected during the 8 day chilled storage, while a significant increase was recorded for the control samples. The results correspond to literature findings reporting negative effects of rinsing on visual quality attributes of green leafy vegetables, including increased yellow discolouration (de-greening), colour heterogeneity as well as loss of phytonutrients, particularly when combined with sanitizer [47,48]. Moreover, water rinsing of rucola has been shown to accelerate the development of off-odor when compared to unwashed samples, regardless of packaging types used during the subsequent cold storage [49].

PAW preservation efficacy on strawberries [13], Chinese bayberries [50], and fresh-cut celery and radicchio leaves [51] has been attributed to the combined action of high oxidation reduction potential and low pH. Although PAW antimicrobial efficacy on strawberries, celery and radicchio has been shown to depend on both the CP activation time for PAW generation and the PAW exposure time on the product, no dose-dependent relationship between PAW treatment and fruit decay was observed for Chinese bayberries, reduced by 50% in all PAW-treated samples after 8 days at 3 °C. On the other hand, while no significant change in colour attributes was found on PAW-treated strawberries and celery after 4 days at 20 °C and 70% relative humidity (R.H), and 5 days at 4 °C and 80% R.H., respectively, a significant decrease of the chroma colour parameter during storage (5 days at 4 °C and 80% R.H.) was observed in PAW-treated radicchio leaves. On the other hand, PAW-treatment promoted higher colour index in Chinese bayberries during 8 days at 3 °C. Xu et al. (2016) have also reported that free oxygen radicals in PAW inhibited browning enzymatic activity in fresh bottom mushrooms, resulting in enhanced quality and shelf-life [14]. Thus, different chemical composition and porous structure (affecting bacterial cell distribution and PAW penetration) of fresh produce might influence PAW antimicrobial efficacy and ability for quality retention [51].

According to the Regulation (EC) 2073/2005 defining mandatory microbiological criteria for foodstuff, the bacterial flora normally present in ready-to-eat vegetables is likely to reach relatively high levels (10^6^–10^7^ CFU/g). Likewise, Ragaert et al. (2007) [52] have reported that evident organoleptic alterations occur in vegetables when total bacterial counts reach 7–8 log CFU/g. In the present study, microbial levels on PAW-treated samples after 8 days of storage at 4 °C reached about 5 log CFU/g, which leaves a relative margin for shelf-life extension beyond current cut-offs at retailers.

## 4. Conclusions

In the present study, a remarkable dependency between PAW chemistry and CP operating parameters (plasma power and exposure time) has been demonstrated, with lower pH values and more pronounced RNS levels being achieved at the most severe conditions for CP exposure time and power. Moreover, at least 2 weeks of chilled storage capacity have been confirmed, which will confer a more flexible operational margin to food producers to cope with the volatile supply and demand. Finally, the potential of PAW for microbial disinfection, quality retention (colour) and shelf-life extension has been demonstrated for baby spinach leaves after 8 days of (air) storage at 4 °C, which will also contribute to reduced industry and household food waste generation. Thus, PAW stands as a promising alternative to traditional sanitisers applied in the fresh produce industry.

## Figures and Tables

**Figure 1 foods-08-00692-f001:**
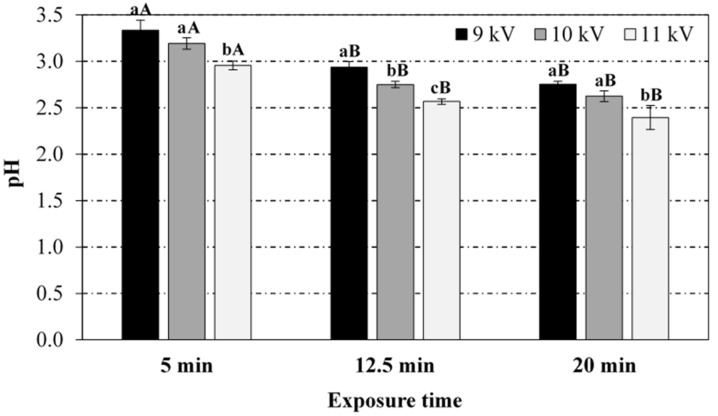
Effect of the voltage peak-to-peak and exposure time to the plasma source on the pH values of the plasma activated water (PAW). For the same exposure time (5, 12.5 or 20 min), bars bearing different lowercase letters are significantly different (*p* ≤ 0.05). For the same voltage peak-to-peak (9, 10 or 11 kV), bars bearing different uppercase letters are significantly different (*p* ≤ 0.05).

**Figure 2 foods-08-00692-f002:**
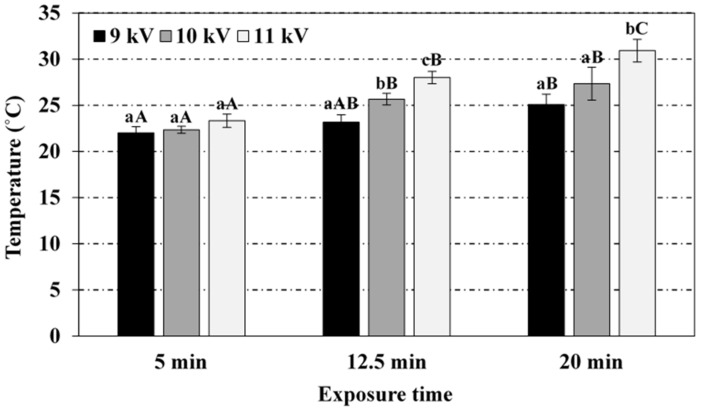
Effect of the voltage peak-to-peak and exposure time to the plasma source on the temperature of the PAW. For the same exposure time (5, 12.5 or 20 min), bars bearing different lowercase letters are significantly different (*p* ≤ 0.05). For the same voltage peak-to-peak (9, 10 or 11 kV), bars bearing different uppercase letters are significantly different (*p* ≤ 0.05).

**Figure 3 foods-08-00692-f003:**
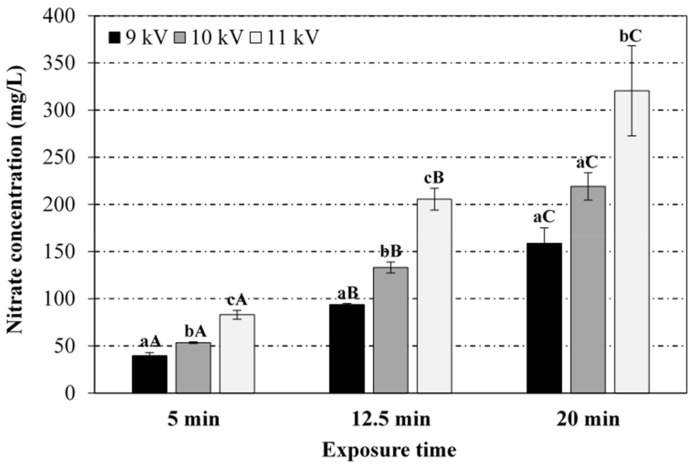
Effect of the voltage peak-to-peak and exposure time to the plasma source on the nitrate levels in the PAW. For the same exposure time (5, 12.5 or 20 min), bars bearing different lowercase letters are significantly different (*p* ≤ 0.05). For the same voltage peak-to-peak (9, 10 or 11 kV), bars bearing different uppercase letters are significantly different (*p* ≤ 0.05).

**Figure 4 foods-08-00692-f004:**
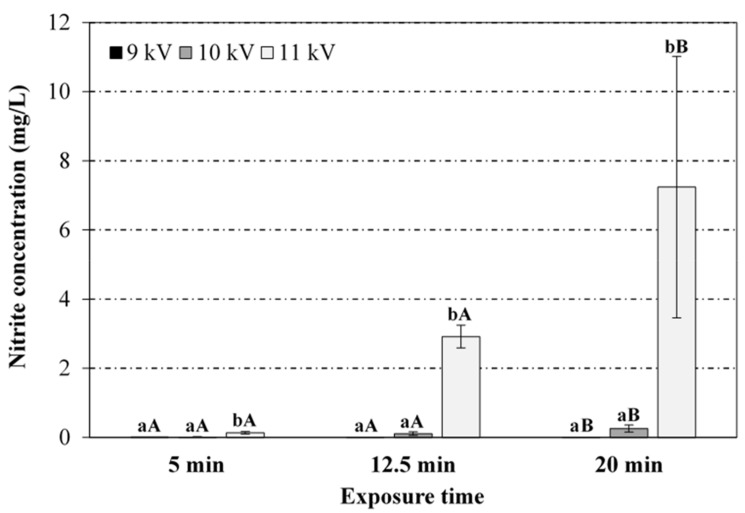
Effect of the voltage peak-to-peak and exposure time to the plasma source on the nitrite levels in the PAW. For the same exposure time (5, 12.5 or 20 min), bars bearing different lowercase letters are significantly different (*p* ≤ 0.05). For the same voltage peak-to-peak (9, 10 or 11 kV), bars bearing different uppercase letters are significantly different (*p* ≤ 0.05).

**Figure 5 foods-08-00692-f005:**
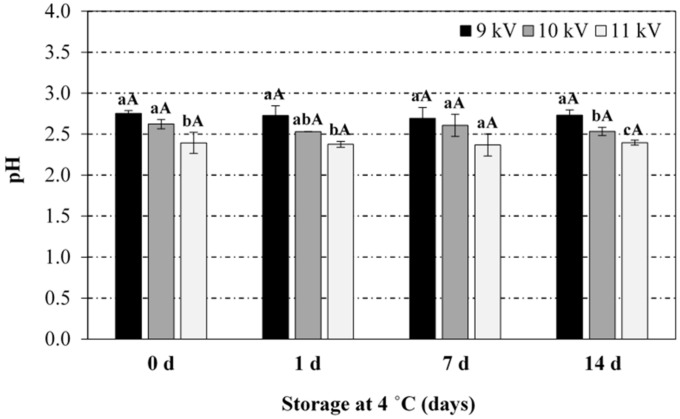
Stability of the PAW (pH) during storage at 4 °C as a function of the voltage peak-to-peak during the PAW generation (20 min treatment). For the same storage time (0, 1, 7 or 14 days), bars bearing different lowercase letters are significantly different (*p* ≤ 0.05). For the same voltage peak-to-peak (9, 10 or 11 kV), bars bearing different uppercase letters are significantly different (*p* ≤ 0.05).

**Figure 6 foods-08-00692-f006:**
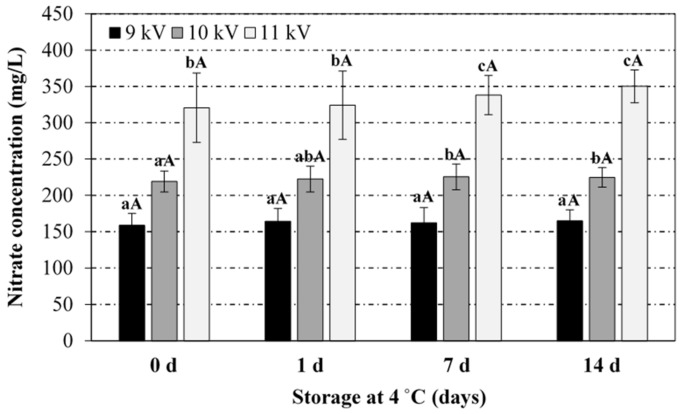
Stability of the PAW (nitrates) during storage at 4 °C as a function of the voltage peak-to-peak during the PAW generation (20 min treatment). For the same storage time (0, 1, 7 or 14 days), bars bearing different lowercase letters are significantly different (*p* ≤ 0.05). For the same voltage peak-to-peak (9, 10 or 11 kV), bars bearing different uppercase letters are significantly different (*p* ≤ 0.05).

**Figure 7 foods-08-00692-f007:**
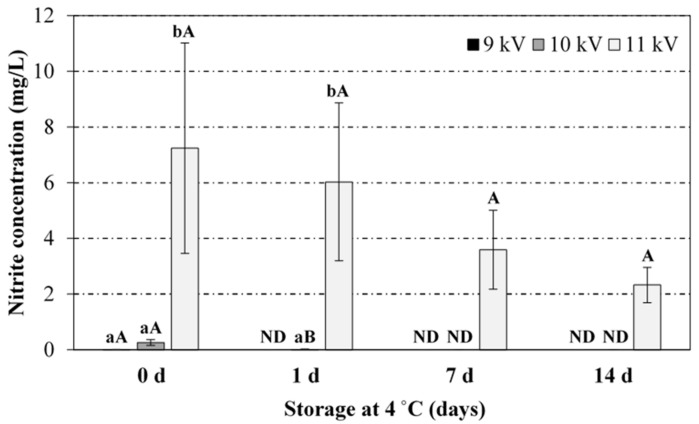
Stability of the PAW (nitrites) during storage at 4 °C as a function of the voltage peak-to-peak during the PAW generation (20 min treatment). For the same storage time (0, 1, 7 or 14 days), bars bearing different lowercase letters are significantly different (*p* ≤ 0.05). For the same voltage peak-to-peak (9, 10 or 11 kV), bars bearing different uppercase letters are significantly different (*p* ≤ 0.05). ND: No detectable concentration.

**Figure 8 foods-08-00692-f008:**
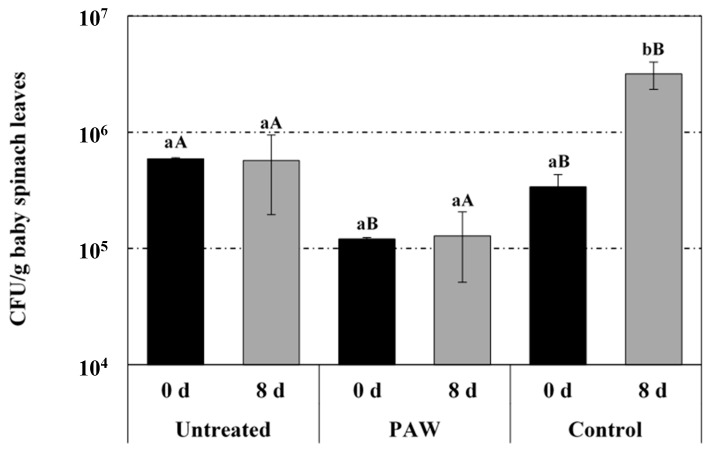
Antimicrobial efficacy of PAW on baby spinach leaves (untreated samples, PAW-treated samples and control samples treated with distilled water) right after treatment (0 day) and after 8 days of storage at 4 °C (8 days). For the same treatment (untreated, PAW or control), bars bearing different lowercase letters are significantly different (*p* ≤ 0.05). For the same storage time (0 or 8 days), bars bearing different uppercase letters are significantly different (*p* ≤ 0.05).

**Figure 9 foods-08-00692-f009:**
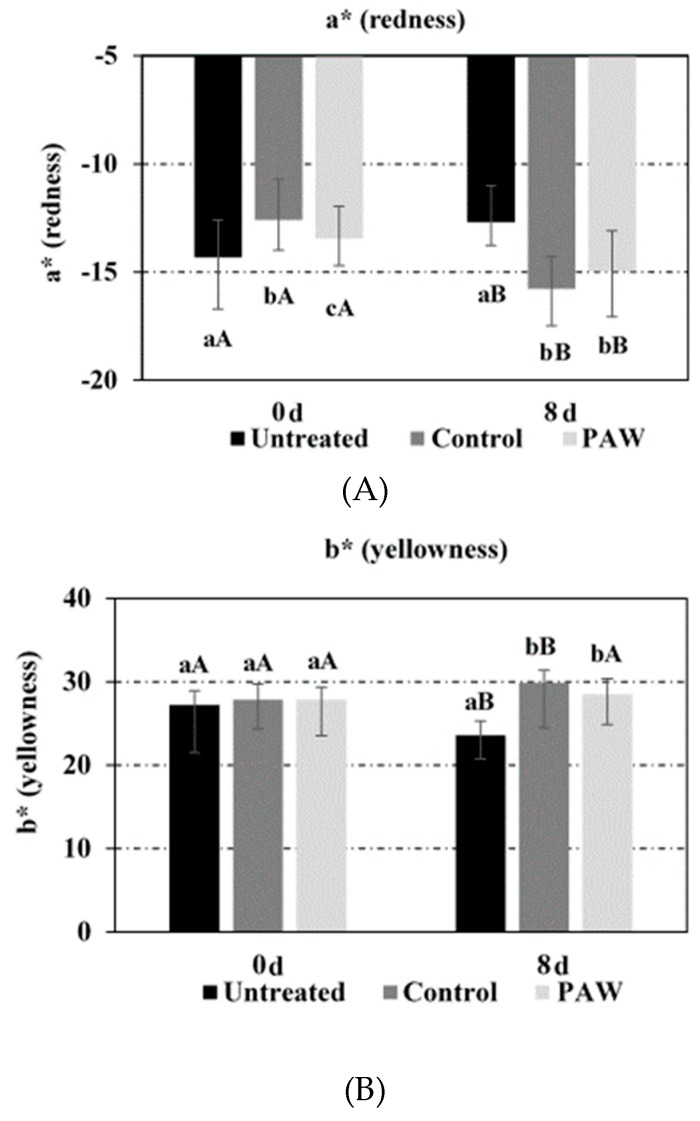
Effect of PAW on lightness (L* value) (**A**), redness (a* value) (**B**) and yellowness (b* value) (**C**) of baby spinach leaves (untreated samples, PAW-treated samples and control samples treated with distilled water) right after treatment (0 day) and after 8 days of storage at 4 °C (8 days). For the same treatment (untreated, PAW or control), bars bearing different lowercase letters are significantly different (*p* ≤ 0.05). For the same storage time (0 or 8 days), bars bearing different uppercase letters are significantly different (*p* ≤ 0.05).

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
