# Peer review of "Towards the Next-Generation Disinfectant: Composition, Storability and Preservation Potential of Plasma Activated Water on Baby Spinach Leaves"

_foods, 2019, doi:10.3390/foods8120692_

Round 1

Reviewer 1 Report

The work is of great interest considering the existing body of knowledge regarding real PAW applications. It is a new experience that can be very useful to other researchers working with PAW as a disinfectant.
Certainly, at present, there is not much research that relates the plasma parameters for PAW generation, the reactive species generated and the disinfectant capacities on real matrices. This study is one of them.
It is well written. It has an acceptable number of tests and characterizations. However, there are certain aspects that if resolved would improve its quality:

It is demonstrated how plasma treatment parameters (time and power) influence pH, NO2 and NO3. But later it does not relate the bactericidal capacity with the chemistry of PAW. It would be interesting if more than one PAW was applied to the final product (only the PAW applies with greater time and power); in order to establish relationships between bactericidal effect, NO2 and NO3. On a couple of occasions it refers to the high ORP produced by plasma promotes antibacterial conservation and mitigates "yellowing". However, it does not show ORP measurement data in the PAWs generated. Do they all have a similar ORP?

Reviewer 2 Report

This manuscript discusses the use of plasma activated water (PAW) as a new-generation disinfectant. The research is very thorough and the text is written in a good scientific way. This reader has however observed question/remarks that should be resolved.

Introduction

Page 2, line 50.

There are some articles that report alkaline values of pH after plasma exposure using, for example, N2 as plasma gas or buffer solutions. Could you rewrite this statement?

Page 2, line 69.

A square bracket is missing, please, check it.

MATERIALS AND METHODS

Page 3, line 162.

What is the reason why you study the PAW stability just for 20 min-treatment PAW?

Please, you should justify this.

Page 4, line 185.

Why have you used these incubated conditions (37 ºC for 48 h) and not others. Certainly, in real life this product is not subjected to these extreme conditions.

RESULTS AND DISCUSSION

Page 4, line 211.

You declare that the absence of hydrogen peroxide has been attributed to the rapid decomposition by nitrites. In this sense, do you think that the quantity of nitrites you has measured  is fewer than the one actually generated by plasma treatment?

Page 4, line 214.

What do you refer as “remote PAW generation”?

Page 6, Line 256.

If you believe that you have not detected H2O2 because of the reaction of this compound with nitrites under acidic conditions, what could be the reason why the levels of nitrates and nitrites are similar when the initial pH of the distilled water is alkaline?

Page 8, line 324.

Why do you store the PAW at 4ºC? If somebody thinks in an industrial application, it is clear that you should store vegetables in chilled storage, but I think that maybe PAW could be store at room temperature. What about this?

Page 10, line 349.

Please, you should reference this study.

Page 11, line 379 and figure 8.

You achieve 1 log reduction in total bacterial count using PAW. However, do you think it is a very promising result?

Page 14, line 424.

Have you measure the ORP of the PAWs you use in this article? Are they in relation with the obtained results?

Maybe, it would give added value to this article.

Reviewer 3 Report

The paper deals with a very important problem, however few crucial points should be clarified before the paper's acceptance.

The plasma activated water produced by the authors contains only nitrite and nitrate as long-lived species, and lacks of hydrogen peroxide, which is considered the main species contributing to the disinfection. Furthermore, during storage also the short-lived species created during plasma treatment are lost. Therefore, the main question is, what is the difference between the plasma activated water containing only nitrite and nitrate and a solution produced from nitric acid.

As a comment for Section 3.2 and the conclusions drawn here in lines 365-367. The ageing of PAW (storage stability) has been recently investigated into details by Kutasi et al. 2019 Plasma Sources Sci. Technol. 28 095010, and the possibilities offered by different plasma systems have been also reviewed. It has been shown that ageing of PAW depends on the initial concentration of the created species, governed by the reactions also detailed in Lukes et al. 2014 PlasmaSources Sci. Technol.23015019. Indeed, from practical point of view the stability depends on the PAW chemistry and generation conditions, but from the chemistry point of view it depends on the initial species concentrations, which also define the pH of the PAW. Related to this, in line 271, it is not the nitrite levels that are attributed to the higher pH values, but vice versa. This would become more clear also for the reader if the authors detail the species creation reactions (related to the text in lines 226-238, and also specify the RONS they refer to) as reviewed in the publications mentioned above.

Concerning the plasma system used. From the application point of view it would be interesting if the authors give the input power and energy density, here energy density in mol/kJ would mean the energy consumed for the creation of 1 mol of total active species. It is also not totally clear what the authors mean by the PAW being generated with a surface dielectric barrier discharge electrode? Is it a SDBD that is located over the water surface? Why would call it electrode?

It is also not clear in line 308-309 the correlation between the liquid volume and gap distance.

Round 2

Reviewer 3 Report

The authors improved the manuscript according to the Referee comments.